# *N*-Chlorotaurine Reduces the Lung and Systemic Inflammation in LPS-Induced Pneumonia in High Fat Diet-Induced Obese Mice

**DOI:** 10.3390/metabo12040349

**Published:** 2022-04-14

**Authors:** Nguyen Khanh Hoang, Eiji Maegawa, Shigeru Murakami, Stephen W. Schaffer, Takashi Ito

**Affiliations:** 1Faculty of Bioscience and Biotechnology, Fukui Prefectural University, Eiheiji 910-1195, Japan; s2073003@g.fpu.ac.jp (N.K.H.); s1821045@g.fpu.ac.jp (E.M.); murakami@fpu.ac.jp (S.M.); 2Department of Pharmacology, College of Medicine, University of South Alabama, Mobile, AL 36688, USA; sschaffe@southalabama.edu

**Keywords:** taurine, *N*-chlorotaurine, lung inflammation, cytokine, muscle wasting, atrophy

## Abstract

Lung infection can evoke pulmonary and systemic inflammation, which is associated with systemic severe symptoms, such as skeletal muscle wasting. While *N*-chlorotaurine (also known as taurine chloramine; TauCl) has anti-inflammatory effects in cells, its effects against pulmonary and systemic inflammation after lung infection has not been elucidated. In the present study, we evaluated the anti-inflammatory effect of the taurine derivative, TauCl against *Escherichia coli*-derived lipopolysaccharide (LPS)-induced pneumonia in obese mice maintained on a high fat diet. In this study, TauCl was injected intraperitoneally 1 h before intratracheal LPS administration. While body weight was decreased by 7.5% after LPS administration, TauCl treatment suppressed body weight loss. TauCl also attenuated the increase in lung weight due to lung edema. While LPS-induced acute pneumonia caused an increase in cytokine/chemokine mRNA expression, including that of IL-1β, -6, TNF-α, MCP-1, TauCl treatment attenuated IL-6, and TNF-alpha expression, but not IL-1β and MCP-1. TauCl treatment partly attenuated the elevation of the serum cytokines. Furthermore, TauCl treatment alleviated skeletal muscle wasting. Importantly, LPS-induced expression of Atrogin-1, MuRF1 and IκB, direct or indirect targets for NFκB, were suppressed by TauCl treatment. These findings suggest that intraperitoneal TauCl treatment attenuates acute pneumonia-related pulmonary and systemic inflammation, including muscle wasting, in vivo.

## 1. Introduction

When an organism is invaded by a pathogen such as bacteria and virus, the innate immune system, including neutrophils and macrophages, is activated to secrete antimicrobial substances and phagocytose the pathogen. Cytokines are secreted to activate the surrounding immune system, eliminate the pathogen and protect the organism. However, a strong immune response can lead to excessive production of cytokines not only from immune cells but also from non-immune cells, such as fibroblasts and vascular endothelial cells, a phenomenon known as a cytokine storm [1]. Cytokine storms can be triggered not only by pathogens, but by cancer, autoimmune diseases, and immunotherapies, causing a lethal systemic immune response due to elevated circulating cytokines. Thus, the activated immune system attacks its own tissues rather than pathogens, resulting in multiple organ failure. 

Taurine is a beta-amino acid found in high concentrations in mammalian tissues [2,3]. Taurine exerts a variety of biological actions, including antioxidation, modulation of ion movement, osmoregulation, and stabilization of macromolecules etc. [4,5] Moreover, treatment of taurine benefits many kinds of pathologies [6,7]. Concerning lung inflammation, previous studies have demonstrated that oral taurine treatment reduces lung inflammation in bleomycin-induced lung injury [8].

Taurine is present in immune system cells, such as macrophages and neutrophils, at levels as high as 35 mM [9]. When an organism is infected by pathogens, immune cells produce hypochlorous acid to kill the pathogens, while taurine reacts with excessive hypochlorous acid to produce *N*-chlorotaurine (also known as taurine chloramine; TauCl), which reduces the high levels of hypochlorous acid and thus its toxicity to surrounding host cells. TauCl also has a broad spectrum of pathogenic microbial (viral, fungal, and bacterial) killing effects [10,11,12,13] and has been reported to inhibit excessive inflammatory reactions through suppression of the production of inflammatory cytokines [14]. In in vitro experiments, TauCl reduces the expression of prostaglandin E2, nitric oxide (NO), and inflammatory cytokines, such as IL-1, -6, -8 and TNF-α [14]. These effects on the regulation of inflammatory cytokine expression have been confirmed not only in immune cells but also in adipocytes and synovial fibroblasts of bone tissue [15,16]. The mechanism underlying the anti-inflammatory action of TauCl is partly associated with the modulation of nuclear factor-κB (NF-κB) [17]. While these results suggest that TauCl may be useful against the excessive production of cytokines that are thought to be involved in the severity of pneumonia, including bacterial infection disease, there is little information about the in vivo effects of TauCl on excessive inflammation.

Lipopolysaccharide (LPS) is a component of the outer membrane of Gram-negative bacteria, which can induce inflammatory responses and is commonly used to study pulmonary inflammation and acute respiratory disease in mice [18]. It binds to cellular Toll-like receptor 4 (TLR4) and induces cytokine expression through NF-κB activation. In the present study, we investigated the effect of TauCl against LPS-induced acute pulmonary/systemic inflammation in obese mice. Moreover, we also evaluated its effect against skeletal muscle atrophy, one of the important systemic reactions of LPS.

## 2. Results

### 2.1. Effects on Lung Inflammation

In our preliminary study, we found that mice maintained on a high fat diet to induce obesity showed more severe weight loss and lung inflammation after LPS administration than normal young mice. Indeed, normal young mice, but not high fat diet-induced obese mice, may start to recover within 2 days after intratracheal LPS administration (Nguyen et al. 2022 in press). Therefore, in the present study, LPS was administered to obese mice, which were fed a high fat diet for 10 weeks to induce obesity and lung inflammation; in some animals the effect of TauCl was examined. Weight loss was significantly suppressed by the administration of TauCl (Figure 1A). At autopsy 2 days later, LPS administration had resulted in a significant increase in lung weight, confirming the site of inflammation (Figure 1B,C). TauCl treatment slowed the inflammatory response in lung tissue, with the increase in lung weight being predominantly suppressed, indicating that edema and immune cell infiltration in the lung are diminished by TauCl treatment. However, TauCl minimized the decrease in spleen weight induced by LPS.

### 2.2. Effect of TauCl on the LPS-Induced Cytokine mRNA Expression in Lung

To explore the effect of TauCl on cytokine expression induced by intratracheal LPS administration, RNA purified from lungs of mice treated with LPS with or without TauCl were analyzed. As shown in Figure 2, cytokines were elevated after LPS injection, but TauCl predominantly decreased the expression of IL-6 and TNF-α (Figure 2). Meanwhile the gene levels of IL-1β and inflammasomes were upregulated by LPS injection but not suppressed by TauCl.

### 2.3. Effects of TauCl on Systemic Cytokine Level in LPS-Treated Mice

To determine the effect of TauCl on the circulation of cytokines of pneumonia model animals, serum levels of cytokines, including, IL-6, and TNF-α, were measured (Figure 3). TauCl significantly suppressed the increase in serum TNF-α levels, while tending to lower IL-6 levels although the effect was not statistically significant.

### 2.4. Effects of TauCl against Muscle Wasting

Elevated levels of circulating cytokines cause several systemic reactions, including muscle wasting [19]. As shown in Figure 1A, TauCl attenuates the loss of body weight after intratracheal LPS administration, implying that TauCl alleviates muscle wasting. We further investigated whether TauCl treatment contributes to the inhibition of muscle wasting in tibialis anterior muscle. Histological examination showed that myofiber size was decreased within 2 days of LPS administration and that TauCl significantly suppressed the decrease in cross sectional area of the fibers (Figure 4). In addition, the expression of Atrogin-1 and MurF1, which are markers of skeletal muscle atrophy, increase in response to LPS administration. By suppressing their expression (Figure 5A,B), TauCl alleviates skeletal muscle wasting induced by intratracheal LPS injection.

Finally, to investigate whether NF-κB is involved in the suppression of muscle atrophy by TauCl, the mRNA expression of IκBα, a product of NF-κB, was analyzed in muscles from LPS-injected mice treated with or without TauCl. While intratracheal LPS injection increased IκBα mRNA of skeletal muscle, TauCl treatment suppressed them (Figure 5C).

## 3. Discussion

While accumulating evidence from in vitro studies suggest that TauCl alleviates inflammation in many types of inflammatory cells, there have been no studies on pulmonary inflammation and the systemic cytokine storm in animals. In the present study, we investigated the effects of intraperitoneal administration of TauCl to obese mice that had previously been treated with LPS to produce a pneumonia-like condition. TauCl treatment suppressed LPS-induced lung edema and attenuated the production of pro-inflammatory cytokines in the lung of those animals, indicating that TauCl suppressed the excessive immune response in lungs. In addition, systemic cytokine elevation induced by intratracheal injection of LPS was partially diminished. These results suggest that TauCl has an inhibitory effect on the severity of inflammation induced by infection-induced pneumonia. In addition, TauCl suppressed the decline in spleen weight, indicating that TauCl may affect the distribution and function of immune cells. Further studies are required to determine how TauCl regulates immune cells in their inflammatory state.

An increase in systemic cytokines causes a variety of systematic symptoms. In the present study, we demonstrated that TauCl treatment alleviated muscle wasting in the LPS-induced lung inflammation model. This anti-wasting role may be caused in part by suppression of circulating inflammatory cytokines. Both IL-6 and TNFα can activate the catabolic signaling pathway in skeletal muscle leading to muscle wasting [20]. Previous in vitro study revealed that TauCl inhibits the cellular response against cytokines, including TNFα and IL-1β [14], implying that TauCl directly prevents the action of cytokines in skeletal muscle. In contrast, a recent study reported that LPS administered according to the intratracheal method can migrate into the blood [21]. Hence, it is possible that TauCl blocks the systemic actions of LPS. Other potential mechanisms may be involved in the anti-wasting actions of TauCl in this model. In the present study, we observed that TauCl suppressed the expression of atrogenes, such as Atrogin-1 and MuRF1, as well as reduced the levels of IκB. MuRF1 and IκBα are the direct downstream targets of NF-κB [20,22] while Atrogin-1 is regulated by NF-κB via nuclear phosphatase SCP4, which is another target of NF-κB [23]. Therefore, these observations suggest a direct action of TauCl on skeletal muscle through the inhibition of NF-κB, which may partly involve an anti-muscle wasting effect.

It has previously been shown that young (8 to 10-week-old) mice administered LPS lose weight and their lungs accumulate fluid (Nguyen et al. 2022 in press). However, weight loss and lung edema were more severe in obese mice. Since adipose tissue also produces cytokines and is the cause of chronic inflammation in obesity, it is possible that LPS administration amplifies the inflammatory response. In this regard it is possible that TauCl can inhibit cytokine production in macrophages in adipose tissue [24]. The results in this study need to be further studied, as TauCl might not only suppress excessive cytokine production by acting on immune cells, but also affect the production of cytokines by adipose tissue macrophage.

In the present study, administration of TauCl was intraperitoneal route, since TauCl is considered to be unstable due to the elimination by anti-oxidation mechanism, such as glutathione (GSH) and ascorbate, or the other amino acids [9,14,25,26]. Yet, we observed that some acute inflammatory events were partially suppressed by TauCl; indeed, some of the injected TauCl may have been transferred to the tissues or may have been dechlorinated to taurine by an anti-oxidation mechanism in body fluid. Since the method of measuring TauCl from crude samples, such as blood and tissues, has not been established, we could not clarify the pharmacokinetics of TauCl in the body. An important limitation of this study is that our research could not clarify how much TauCl itself is actioned in cells. Taurine, which were converted from TauCl, might partly contribute to reduce lung or systemic inflammation. Indeed, it has been reported that systemic (intraperitoneal or intravenous) administration of taurine alleviated inflammatory responses in vivo studies [27,28].

Kwaśny-Krochin et al. reported that a single dose of TauCl in collagen-induced arthritis delayed the onset of inflammation, but did not diminish the severity of arthritis and the onset of cytokine production at the end of experiments [29]. Meanwhile, Wang et al. observed that continuous subcutaneous injections of TauCl attenuated arthritic symptoms and synovial inflammation in collagen-induced arthritic mice [30]. Thus, it is logical to conclude that sustained administration of TauCl may be more effective than a single administration of TauCl in suppressing the production of pro-inflammatory cytokines in the lung of LPS-treated mice with acute lung inflammation.

A novel type of coronavirus infection (COVID-19) has been spreading around the world since 2020, and by the end of 2021, about 5.4 million people have lost their lives (Available online: https://covid19.who.int/ (accessed on 29 December 2021). In patients with COVID-19, there is a correlation between the levels of cytokines in the blood and the severity of the disease, suggesting that excessive activation of the immune response contributes to the severity of the disease [31]. Therefore, it is desirable to develop therapies to limit the immune response, as well as antiviral drugs against the causative virus. Recent studies have revealed that TauCl can inactivate SARS-CoV-2 in vitro [32]. In addition to its antimicrobial role, the present findings raise the possibility that TauCl treatment might alleviate the severe symptoms associated with acute pneumonia, including COVID-19.

## 4. Materials and Methods

### 4.1. TauCl Synthesis

TauCl was synthesized as previously described [29]. In brief, 20 mM sodium hypochlorite solution (Nacalai tesque, Kyoto, Japan) was added to 24 mM taurine (Nacalai tesque) dissolved in 50 mM phosphate buffer (pH 7.4). Production of TauCl was spectrophotometrically confirmed by UV absorption spectra (220–340 nm, Amax = 252 nm). The concentration of TauCl was determined using the molar extinction coefficient (ε252 nm = 415 M^−1^ cm^−1^) [17].

### 4.2. Animal Care

All experimental procedures were approved by the Institutional Animal Care and Use Committee of the Fukui Prefectural University. Male C57BL/6J mice (6-week-old, Japan Crea) were used for this study. Mice were fed 60% fat-containing chow (Oriental Yeast, Tokyo, Japan) for 10 weeks to induce obesity [33]. The mice had access to water ad libitum, and were maintained on a 12-h light/dark cycle.

### 4.3. LPS-Induced Pneumonia Model

Pneumonia was induced by intratracheal injection of a LPS solution (O-111: 7 mg/kg body weight, Fuji Film Co. Ltd., Tokyo, Japan) under anesthesia with a combination of medetomidine hydrochloride (0.3 mg/kg, Nippon Zenyaku Kogyo Co., Ltd., Fukushima, Japan), midazolam (4.0 mg/kg, Astellas Pharma Inc., Tokyo, Japan) and butorphanol (5.0 mg/kg, Meiji Seika Pharma Co., Ltd., Tokyo, Japan), as previously described [34]. An equivalent amount of vehicle (PBS) was injected for the control mice. After injection, atipamezole (0.3 mg/kg, Nippon Zenyaku Kogyo Co., Ltd.) was treated to recover from anesthesia. One hour before LPS injection, the mice were intraperitoneally administered a TauCl solution (5 mM, 250 μL) or PBS. Two days after LPS injection, mice were euthanized, and whole blood was immediately collected by retro-orbital bleeding. Then, tissues were collected, immediately frozen in liquid nitrogen and stored at −80 °C until use.

### 4.4. mRNA Measurement

Total RNA was extracted from mouse tissues by using Sepasol (Nacalai tesque) according to the manufacturer’s protocol. cDNA was generated from total RNA by reverse transcription with Rever Tra Ace (Toyobo, Japan). Quantitative RT-PCR analyses were performed by using qTOWER^3^ (Analytik Jena GmBH, Jena, Germany) with KAPA SYBR Fast qPCR Kit (Kapa Biosystems, Inc., Wilmington, MA, USA). The sequence for primers were listed in Table 1. Some of the primers were previously described [35] (Khanh2022 in press). GAPDH is used as an internal control. Data were analyzed using the ∆∆Ct method.

### 4.5. Plasma Cytokine Assay

Serum cytokine concentrations (IL-6 and TNF-α) were measured by ELISA by using the Mouse IL-6 Quantikine ELISA Kit (R&D Systems, Inc., Abingdon, UK), Mouse TNF-αELISA Kit (Fujifilm, Osaka, Japan) respectively, according to the manufacturer’s protocol.

### 4.6. Tissue Section and Immunostaining

Sections from frozen tissues were cut by cryostat (Leica Microsystems, Wetzlar, Germany). For detection of Laminin, the frozen section was immunostained by using anti-laminin antibody (ab80, Abcam, Cambridge, UK; 1:100) and Alexa Fluor 488-conjugated second antibody (Life Technologies; 1:400) with Can Get Signal Immunostain according to manufacturer’s protocol (Toyobo, Osaka, Osaka). Images were acquired with microscopes (BZ-9000, Keyence, Osaka, Osaka) equipped with imaging software (BZ-II, Keyence). Cross-sectional area was calculated by using ImageJ software (version: 1.53e).

### 4.7. Statistical Analysis

Data are presented as the means  ±  standard deviation. Microsoft Excel and R software packages were used for statistical analysis of the data. Data were tested for normal distribution using the Shapiro–Wilk normality test. The non-parametric Mann–Whitney U-test and the parametric independent *t*-test and Tukey-Kramer’s *t*-test were used to determine statistical significance between groups. The Grubbs test was used to detect and reject outliers. Differences were considered statistically significant when the calculated *p*-value was less than 0.05.

## Figures and Tables

**Figure 1 metabolites-12-00349-f001:**
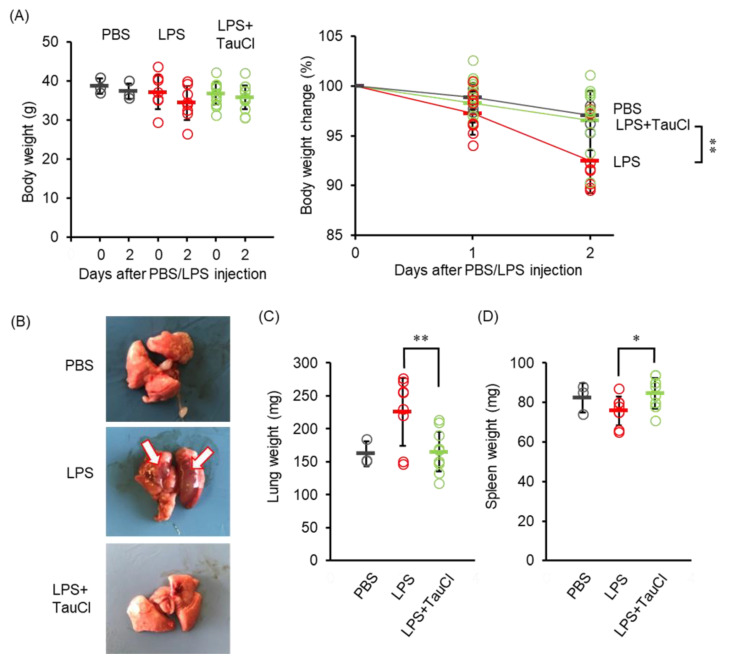
Effect of TauCl on lung inflammation induced by intratracheal LPS. (**A**) The changes in body weight after intratracheal PBS or LPS injection with TauCl. Body weight was monitored after LPS injection. (**B**) Representative lung images isolated from mice at 2 days after LPS injection. White arrows indicate the inflammation site. (**C**,**D**) The weight of lungs (**C**) and spleens (**D**) was shown. TauCl was intraperitoneally pretreated 1 h prior to LPS injection in the TauCl-treated group (LPS + TauCl). Values shown represent means ± SD. *n* = 3 (PBS), 9 (LPS), and 11 (LPS + TauCl). * *p* < 0.05, ** *p* < 0.01 (Tukey-Kramer’s *t*-test).

**Figure 2 metabolites-12-00349-f002:**
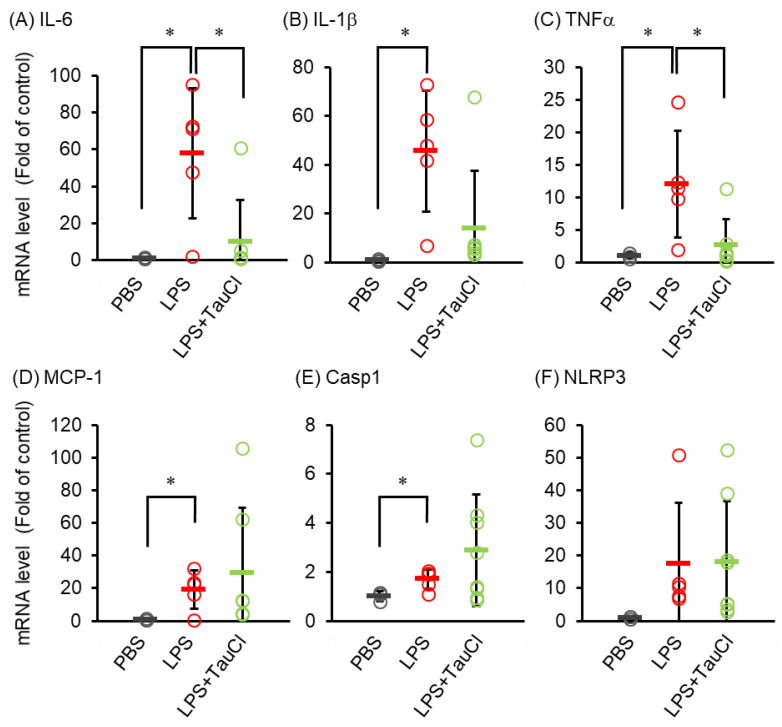
Effect of TauCl on LPS-induced expression of cytokine/chemokine mRNA in the lung. The lungs were isolated from mice 2 days after PBS or LPS injection. TauCl was intraperitoneally pretreated 1 h prior to LPS injection (LPS + TauCl). The expression of IL-6 (**A**), IL-1β (**B**), TNFα (**C**), MCP-1 (**D**), Casp1 (**E**) and NLRP3 (**F**) were measured by qPCR. The expression level was normalized by the expression level of GAPDH. Values are shown fold of control (PBS). *n* = 3 (PBS), 8 (LPS), and 11 (LPS + TauCl). * *p* < 0.05 (Tukey–Kramer’s *t*-test for IL-6, IL-1β, MCP-1, Casp1 and Mann-Whitney U-test for TNFα).

**Figure 3 metabolites-12-00349-f003:**
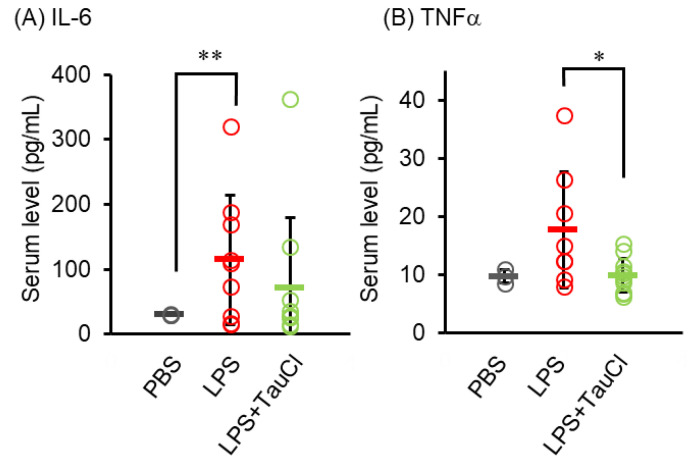
Effect of TauCl and taurine on serum cytokine level. Blood was collected from mice 2 days after PBS or LPS injection. Serum IL-6 (**A**) and TNF-α (**B**) levels were measured by ELISA. Values shown represent means ± SD. *n* = 3 (PBS), 8 (LPS), and 11 (LPS + TauCl). * *p* < 0.05, ** *p* < 0.01 (Tukey–Kramer’s *t*-test).

**Figure 4 metabolites-12-00349-f004:**
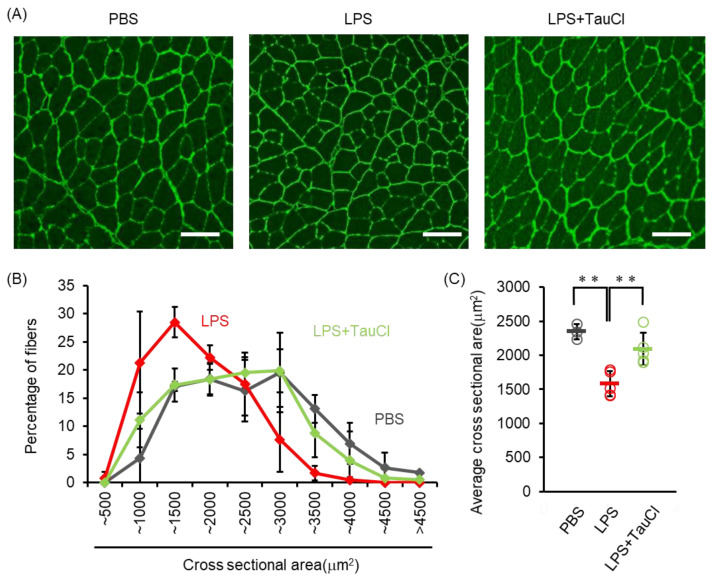
Effect of TauCl on skeletal muscle wasting induced by intratracheal LPS. The skeletal muscles (tibialis anterior muscles) were isolated from mice 2 days after PBS or LPS injection. TauCl was intraperitoneally pretreated 1 h prior to LPS injection (LPS + TauCl). Frozen section of skeletal muscles were immunostained with anti-laminin antibody. (**A**) Representative cross sectional image of skeletal muscle. Scale bars indicate 100 μm. (**B**) Cross sectional area of skeletal muscle fiber was measured from the immunostained images. Values indicate the frequency (percentage) of each size of muscle fiber. (**C**) The average of cross sectional area of skeletal muscle of each mouse was shown. *n* = 3 (PBS), 5 (LPS), and 5 (LPS + TauCl). ** *p* < 0.01 (Tukey–Kramer’s *t*-test).

**Figure 5 metabolites-12-00349-f005:**
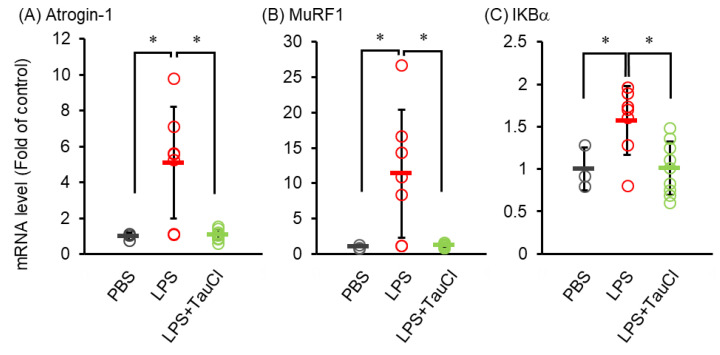
Effect of TauCl on the atrogenes and IKBa expression in skeletal muscle in LPS-treated mice. The tibialis anterior muscles were isolated from mice 2 days after PBS or LPS injection. TauCl was intraperitoneally pretreated 1 h prior to LPS injection (LPS + TauCl). The expression of Atrogin-1 (**A**), MurF1 (**B**) and IkBα (**C**) were measured by qPCR. The expression level was normalized by the expression level of GAPDH. Values are shown fold of control (PBS). *n* = 3 (PBS), 7 (LPS), and 7 (LPS + TauCl). * *p* < 0.05 (Tukey–Kramer’s *t*-test).

**Table 1 metabolites-12-00349-t001:** Primers for quantitative PCR.

Gene	Forward Primer (5′→3′)	Reverse Primer (5′→3′)
IL-1β	CCTTG GGCCT CAAAG GAAAG A	TTGCT TGGGA TCCAC ACTCT CC
IL-6	CACTT CACAA GTCGG AGGCT T	GAATT GCCAT TGCAC AACTC TTTTC
TNF-α	CAAAA TTCGA GTGAC AAGCC TGTA	CACCA CTAGT TGGTT GTCTT TGAGA
MCP-1	CTGTC ATGCT TCTGG GCCTG	GGCGT TAACT GCATC TGGCT GA
Casp1	GCATG CCGTG GAGAG AAACA	ATGGG CCTTC TTAAT GCCAT CAT
NLRP3	GCAGA GCCTA CAGTT GGGTG	CTTCC ACGCC TACCA GGAAA T
Atrogin-1	TTCAG CAGCC TGAAC TACGA	AGTAT CCATG GCGCT CCTTC
MuRF1	GCGTG ACCAC AGAGG GTAAA	CTCTG CGTCC AGAGC GTG
IκBα	TAGCA GTCTT GACGC AGACC	GACAC GTGTG GCCAT TGTAG
GAPDH	GCCGG TGCTG AGTAT GTCGT	CCCTT TTGGC TCCAC CCTT

## Data Availability

Data is contained within the article.

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
