# Peer review of "N-Chlorotaurine Reduces the Lung and Systemic Inflammation in LPS-Induced Pneumonia in High Fat Diet-Induced Obese Mice"

_metabolites, 2022, doi:10.3390/metabo12040349_

Round 1
Reviewer 1 Report
Please see enclosed file.

Author Response
Dear the Editor,
We wish to thank you and the reviewers for helpful comments, which we believe have improved our manuscript. We have revised the manuscript according to the reviewer’s comments, as follows. We believe that this paper is improved well.
Sincerely,
Takashi Ito
#Reviewer 1
Review on the manuscript metabolites-1655025 entitled „Title: Taurine chloramine reduce the lung and systemic inflammation in LPS-induced pneumonia in high fat diet-induced obese mice“
General comments:
This is a highly interesting study in which the authors show reduction of lung and systemic inflammation by intraperitoneal TauCl in LPS-induced pneumonia in mice. The results are promising and confirm down-regulation of proinflammatory cytokines and chemokines and, moreover, decrease of skeletal muscle wasting. The study has been well done and should be published. There is only one major point, which I miss not only in this study, but also in recent others that investigate systemic effects of topically applied TauCl.
We know from a plethora of studies that the amino acid taurine, from which TauCl is formed, has anti-inflammatory effects, too. Moreover, it is known that topically applied TauCl is not distributed systemically (due to immediate reduction by reducing agents in the blood), but loses its oxidative capacity at the site of application after some time. Thereby, it is reduced to taurine again. Naturally, a mixture of chloramines is formed after application of TauCl, from which again reaction products will be formed. From these considerations, it follows that the found systemic effects will not be caused by TauCl itself but by the formed taurine and possibly in addition by any non-oxidizing reaction products.
To obtain an idea if it deals not only with effects of taurine but an additional effect, a parallel control / comparison group with equimolar taurine is decisive. I really wonder why this has not been done in several studies that deal with systemic effects after topical application of TauCl.
I do not make such a control group a condition for publication of this study, but I strongly suggest this for the future. From the data presented, it appears not yet clear if it deals with TauCl or taurine effects. In the present study, these considerations should be mentioned in the discussion. Maybe, the authors have some ideas on this and can speculate which effects could be related to taurine and which might be additional effects.
> I agree his/her comment. Since there are many evidence for the pharmacological effect of taurine in animal phathological model, we investigated the effect of TauCL, but not taurine in the present study. However, as the reviewer indicated, we should have tested taurine as control study.
Largely minor points:
Throughout the manuscript: The greek letters (IL-1 beta, TNF-alpha etc) do frequently not appear in the version for the reviewer. Please correct any typing problems with these letters.
>I apologize for many typos. We checked greek letters.
Title and nomenclature: Unfortunately, the expression “taurine chloramine” has been established by molecular biologists, although it is not completely correct since the expression taurine itself contains the amino group. Therefore the chlor-“amine” in taurine chloramine is redundant. According to the chemists’ nomenclature, N-chlorotaurine is the correct expression. It is very difficult to change such expressions, but I just want to mention it since it is my duty as a reviewer.
> I agree. I changed the title to “… N-chlorotaurine …” as well as I added the “N-chlorotaurine” in the abstract and introduction.
Line 81: Is there any reason why TauCl was administered one hour before LPS injection ? (Does it not work if administered after application of LPS ? If there is a special reason, it should be mentioned here and possibly discussed in the discussion section)
>The reason why TauCl was pretreated before LPS was the previous studies. We will test the effect of post-treatment of TauCl as well as long-term treatment in future.
Line 84: When and how was blood taken from the mice ?
>The method to take blood was added in Method section in the revised manuscript, as follows;
Two days after LPS injection, mice were euthanized, and whole blood was immediately collected by retro-orbital bleeding. Then, tissues were collected, immediately frozen in liquid nitrogen and stored at -80°C until use.
Lines 106-8 (Statistics): The data presentation in the figures indicates non-Gaussian distribution in part. Were all the data normally distributed or should non-parametric tests be used in part (e.g. Mann-Whitney, Kruskal-Wallis) ?
> We checked the Gaussian distribution of the data as the reviewer indicated, and we found some non-Gaussian distributed data. Then, we evaluated the statistical significance again with non-parametric tests. I changed Method section, as follows;
Data are presented as the means ± standard deviation. Microsoft Excel and R software packages were for statistical analysis of the data. Data were tested for normal distribution using the Shapiro-Wilk normality test. The non-parametric Mann-Whitney U-test and the parametric independent t-test and Tukey-Kramer’s t-test were used to determine statistical significance between groups. The Grubbs test was used to detect and reject outliers. Dif-ferences were considered statistically significant when the calculated P-value was less than 0.05.
Lines 198-243: These effects discussed are obviously effects of the formed taurine and / or of other rection products, but very improbably direct effects of TauCl. Please discuss.
Lines 211-217: “... suggest a direct action of TauCl on skeletal muscle ...“
The point is that TauCl does not get there if not injected directly into the muscle.
> I added a discussion about whether TauCl directly acts in the tissues or not in Discussion section, as follows;
An important limitation of this study is that our research could not clarify how much TauCl itself actioned in cells. Taurine which were converted from TauCl might partly con-tribute to reduce lung or systemic inflammation. Indeed, it has been reported that systemic (intraperitoneal or intravenous) administration of taurine alleviated inflammatory re-sponses in vivo studies [32,33].
Line 222: At least according to the abstract, taurine, not TauCl, was used in animal experiments. There is only one sentence indicating inhibition of M1 differentiation by TauCl in vitro. Please correct the sentence of line 222-223 accordingly if this is right.
>I changed the discussion as follows;
In this regard it is possible that TauCl can inhibit cytokine production in macrophages in adipose tissue [29].
Line 227: TauCl is an oxidant, not an antioxidant, and therefore it has oxidizing activity, not antioxidant one. Please correct.
>消しました。
Lines 231-243: Are there any taurine control groups in these studies or well-founded speculations if TauCl has more effect than taurine ?
>According to previous in vitro studies, TauCl, but not taurine, reduces cellular inflammatory responses, as mentioned in the manuscript. Therefore, I concluded the anti-inflammation effect may be due to the direct effect of TauCl in body. However, I found some reports which demonstrated that systemic administration of taurine reduced inflammation (reference 32,33). So as the reviewer indicated, we should test the effect of systemic taurine administration in LPS-induced pneumonia model in future.
Reviewer 2 Report
The research article entitled “Taurine chloramine reduce the lung and systemic inflammation in LPS-induced pneumonia in high fat diet-induced obese mice” by Khanh Hoang Nguyen etal analyze the protective effect of Taurine chloramines on inflammation in LPS-induced pneumonia in mouse. Authors have performed a great study in terms of data and designing of experiments, but typos and sentence construction greatly hampered the readability.
Following are minor comments can be incorporated.
1.Please add some more point about taurine
(Biomolecules 2020, 10, 863; doi:10.3390/biom10060863).
2.Taurine is known for its antioxidant and anti-inflammatory activity. How the effect Taurine chloramines is different from taurine?
3. I think authors should indicate the wavelength range at which Taurine chloramines production was confirmed. How the purity of Taurine chloramines was checked. Name of chemical supplier can be included.
Minor comments-
Please re-write Line no 150-152
Line no 40- please correct body
Please write full form of HClO
Line no 94 TNF- is missing
And many more
Author Response
Dear the Editor,
We wish to thank you and the reviewers for helpful comments, which we believe have improved our manuscript. We have revised the manuscript according to the reviewer’s comments, as follows. We believe that this paper is improved well.
Sincerely,
Takashi Ito
#Reviewer 2
The research article entitled “Taurine chloramine reduce the lung and systemic inflammation in LPS-induced pneumonia in high fat diet-induced obese mice” by Khanh Hoang Nguyen etal analyze the protective effect of Taurine chloramines on inflammation in LPS-induced pneumonia in mouse. Authors have performed a great study in terms of data and designing of experiments, but typos and sentence construction greatly hampered the readability.
Following are minor comments can be incorporated.
1.Please add some more point about taurine (Biomolecules 2020, 10, 863; doi:10.3390/biom10060863).
>I added the introduction about taurine in the revised manuscript.
2.Taurine is known for its antioxidant and anti-inflammatory activity. How the effect Taurine chloramines is different from taurine?
>Accoding to previous in vitro studies, TauCl, but not taurine, reduces inflammation and macrophage differentiation. However, in vivo studies demonstrated that systemic administration of taurine also reduces inflammation. This content was added in the revised manuscript.
- I think authors should indicate the wavelength range at which Taurine chloramines production was confirmed. How the purity of Taurine chloramines was checked. Name of chemical supplier can be included.
>I added the wavelength range for monitoring taurine chloramine in the method section. We could not check the exact purity of Taurine chloramine.
Minor comments-
Please re-write Line no 150-152
> I rewrote this part in the revised manuscript. I apologize for the many typos.
Reviewer 3 Report
I congratulate the authors on presenting such a fine study regarding the protective effects of Taurine chloramine on pulmonary inflammation. It has been shown previously Taurine chloramine reduces lung inflammation see comments below but the strong point of the paper is the measurements regarding skeletal muscle wasting and serum cytokine levels. Even though the paper has many innovative points I do have some concerns regarding the paper.
Major
1) In the introduction authors have missed out on a key publication regarding the effects of Taurine chloramine on pulmonary inflammation. It has been shown previously that bleomycin-induced inflammation in the lungs is reduced with Taurine chloramine (DOI: 10.1007/978-1-4615-0077-3_48).
2) In the methods section you mention that you measured serum Il-1, il-6, and Tnf alpha but the results only show Il-6 and Tnf alpha. Why not il1 ?
3) There seems to be one outlier (LPS+TauCl) group in all the parts in figure 2. Are the results still significant after removing this outlier?
4) I am a bit perplexed about the number of mice included in each group. Why PBS only had 3 mice, LPS had 9 and LPS+TauCl had 11. Do the authors have any special reason for recruiting different numbers of mice in each group.
Minor concerns:
1) it might be during the conversion that special characters like α are missing from a lot of variables. Please check this.
2) Line 40 the spelling of body needs to be corrected
3) Please add how the cross-sectional area of skeletal muscle fibers was measured in the method section.
Author Response
Dear the Editor,
We wish to thank you and the reviewers for helpful comments, which we believe have improved our manuscript. We have revised the manuscript according to the reviewer’s comments, as follows. We believe that this paper is improved well.
Sincerely,
Takashi Ito
#Reviewer 3
I congratulate the authors on presenting such a fine study regarding the protective effects of Taurine chloramine on pulmonary inflammation. It has been shown previously Taurine chloramine reduces lung inflammation see comments below but the strong point of the paper is the measurements regarding skeletal muscle wasting and serum cytokine levels. Even though the paper has many innovative points I do have some concerns regarding the paper.
Major
1) In the introduction authors have missed out on a key publication regarding the effects of Taurine chloramine on pulmonary inflammation. It has been shown previously that bleomycin-induced inflammation in the lungs is reduced with Taurine chloramine (DOI: 10.1007/978-1-4615-0077-3_48).
> I added the indicated article, as follows; In the previous study, oral taurine treatment reduces lung inflammation in bleomy-cin-induced lung injury [8].
This article showed the effect of taurine, but not taurine chloramine.
2) In the methods section you mention that you measured serum Il-1, il-6, and Tnf alpha but the results only show Il-6 and Tnf alpha. Why not il1 ?
> Since ELISA kit for IL-1 did not work for us, we did not show the result for IL-1beta. I forgot to erase it from method section. I appreciate.
3) There seems to be one outlier (LPS+TauCl) group in all the parts in figure 2. Are the results still significant after removing this outlier?
> In this study, we evaluated the statistical significance after removing outliers.
4) I am a bit perplexed about the number of mice included in each group. Why PBS only had 3 mice, LPS had 9 and LPS+TauCl had 11. Do the authors have any special reason for recruiting different numbers of mice in each group.
> Although we initially planned to have 5-6 control (PBS), we failed the intratracheal administration for a few mice, and we reduced the numbers of control. We do not have special reason for it.
Minor concerns:
1) it might be during the conversion that special characters like α are missing from a lot of variables. Please check this.
2) Line 40 the spelling of body needs to be corrected
> I apologize for many typos.
3) Please add how the cross-sectional area of skeletal muscle fibers was measured in the method section.
> I added the method for measurement of cross sectional area.
Round 2
Reviewer 2 Report
I think authors have successfully answers all my queries. The MS can be accepted in current form.